# Influencing factor modeled examination on internet rural logistics talent innovation mechanism based on fuzzy comprehensive evaluation method

**Hui Zhan[1], Xin Zhang[2,3]\*, Haiwen Wang[1]**

**1** School of Economics and Trade, JiLin Engineering Normal University, Changchun, Jilin, China, **2** College of Business, Jiaxing University, Jiaxing, Zhejiang, China, **3** Xinjiang Institute of Technology, Akesu, Xinjiang, China

\* zcxin9@163.com

**Data Availability Statement:** All relevant data are within the manuscript and its Supporting Information files.

## Abstract

In recent years, China's economic development has advanced by leaps and bounds, but the development of China's rural logistics system is still at its primary stage. Some remote areas with inconvenient transportation are still in a state of serious lack or even blank, and due to the high cost of rural logistics delivery services, the rural logistics business of the enterprise also has a profit problem, which limits the development of rural logistics talent innovation to some extent. The purpose of this paper is to study a new influencing factor model of the Internet rural logistics talent innovation mechanism. This paper innovatively proposes countermeasures to improve the innovation of e-commerce practitioners in rural areas. Through research, the author finds that the innovation of rural e-commerce application talents in China is generally low. The key point of the solution lies in how to improve the level of innovation in rural e-commerce application talents. According to the status quo, identify the factors that hinder the innovation and improvement of rural e-commerce application talents. Combined with the great environment of the development of rural e-commerce industry in China, the paper proposes to improve the countermeasures for improving the innovation of rural e-commerce application talents. Improve the current situation of rural e-commerce application talents mediocrity and promote the innovation of rural e-commerce application talents. Fundamentally promote agricultural development and the building of a new socialist countryside. This paper adopts the literature research method based on fuzzy comprehensive evaluation method, system analysis method and the combination of questionnaire survey and interview. Through big data and information science methods for data processing, using a company's Internet rural talent data set to simulate, the results It shows that with the method of this paper, the recognition rate reaches 98%, the speed increases obviously, and it is 20% faster than others.

**Funding:** This work was Supported by Doctoral Research Initiation Funding Project of JiLin Engineering Normal University , Project number: BSSK201803, Project Leader: Hui Zhan; Supported by Zhejiang Soft Science Research Project of China,Project number:2021C35088;Supported by School-level scientific research projects of JiLin Engineering Normal University, Project number: XZD201808; Supported by Program for Innovative Research Team of JiLin Engineering Normal University.

**Competing interests:** The authors have declared that no competing interests exist.

## Introduction

Logistics refers to the planning, implementation and management of raw materials, semi-finished products, finished products or related information from the origin of the goods to the place where the goods are consumed; meet the needs of customers at the lowest cost through transportation, storage and distribution. Today, many countries are contracting providers of logistics to complement the internal distribution of public health systems. This reflects recent major outsourcing initiatives to address major gaps in transportation and logistics [1]. Investigate the application of logistics and supply chain management in practice by investigating the four perspectives of the relationship between logistics and supply chain management, especially at the strategic level. The key factors and inhibitors of success provide a sound basis for realizing the strategic potential of logistics and supply chain management. New insights into the practitioners' perspectives on logistics and supply chain management have been developed and demonstrated, as well as new insights into the factors driving and inhibiting strategic SCM adoption [2]. Big data analytics and the Internet of Things are supporting large logistics companies to improve driver safety, reduce operating costs and reduce environmental impact. This has implications for logistics companies in other parts of the world, where they may not be as widespread, but more widely used by drivers and society [3]. Iran is a country with great potential for logistics activities in the Middle East and links logistics networks in Asia, Europe and Africa. Management and leadership, internationalization and employee competencies are the most important key success factors for Iranian logistics supplier companies [4]. For future research, other monetary and non-monetary factors should be considered, such as land, labor, investment incentives, and local government regulation [5].

Modern logistics is the product of economic globalization and an important service industry to promote economic globalization. The world's modern logistics industry is growing steadily, and Europe, the United States, Japan have become important logistics bases worldwide. The gradual growth of e-commerce sales and the increasing interest of scholars and practitioners in the omni-channel retail industry. Despite growing interest in OC retail, many key themes remain inadequate, including the evolution of retail distribution networks, multi-channel classification planning, the logistics role of stores in the delivery process, and the interaction between different logistics aspects [6]. Road transport company waste reduction practices have had a positive and significant impact on fleet operational efficiency. Road transport companies can use efficiency measurements as a basis for developing future action plans to adopt waste practices [7]. The average potential of small businesses is greater than that of large companies, and investment significantly affects the company's performance. These companies are Asia's growing logistics hub [8]. To determine that a customer can and does participate in logistics; the company may consider transferring some logistics activities to or from customers. A key issue for companies in the transfer process is the need to redesign their SC in transportation, warehousing and production. Customer engagement logistics is a key variable in SC design [9]. The purchasing manager found that the shared responsibility was to positively influence the delivery speed of the supplier, while the logistics manager found that it had a positive impact on the supplier's price performance. In general, the purchasing manager believes that there is a stronger connection between formal and integrated work (contact roles and joint reward systems) and supplier performance, while logistics managers consider this connection to be informal and comprehensive for information exchange and collaboration. Work is more powerful [10].

In order to solve the problem of less research on rural logistics talent innovation mechanism, this paper adopts the method of literature research, system analysis and questionnaire survey and interview based on fuzzy comprehensive evaluation method, and data processing

through big data and information science methods. Using a company's Internet rural talent dataset to simulate, the results show that with this method, the recognition rate reached 98%, the speed increased significantly, 20% faster than others. Through the investigation and analysis of e-commerce practitioners in some rural areas of Hubei Province, the needs and problems of rural e-commerce talents in talent innovation are explored and summarized. This paper innovatively proposes countermeasures to improve the innovation of e-commerce practitioners in rural areas. Through research, the author finds that the innovation of rural e-commerce application talents in China is generally low. The key point of the solution lies in how to improve the level of innovation in rural e-commerce application talents. According to the status quo, identify the factors that hinder the innovation and improvement of rural e-commerce application talents. Combined with the great environment of the development of rural e-commerce industry in China, the paper proposes to improve the countermeasures for improving the innovation of rural e-commerce application talents. Improve the current situation of rural e-commerce application talents mediocrity and promote the innovation of rural e-commerce application talents. Fundamentally promote agricultural development and the building of a new socialist countryside.

The main structure of this paper is as follows: Section 1 introduces overview. This part mainly introduces the background, research purpose and research significance of the thesis, and further determines the research content and research methods and the innovation of the thesis. Section 2 is related theoretical research and research methods. This paper analyzes the definition of rural e-commerce talents and profoundly expounds the connotation of talent innovation. Through literature review of domestic and international research status, it is hoped to find the starting point of research and lay the foundation for the later research. Section 3 introduces the experimental part of the influencing factor model of the Internet rural logistics talent innovation mechanism. The information acquisition and processing capabilities, information expression and communication capabilities, information ethics and security capabilities are described in detail. The index system is determined according to the influencing factors identified in the previous section, and the evaluation model is proposed and constructed. Section 4 introduces the fuzzy comprehensive evaluation method to empirically analyze and evaluate the rural e-commerce in Hubei Province. This chapter starts from the actual national conditions of our country and elaborates from the perspective of rural e-commerce in Hubei Province. It believes that the development situation of agricultural e-commerce talents in China is quite serious, and it needs a perfect promotion strategy to help the rapid growth of agricultural e-commerce talents. Section 5 introduce the main conclusions of this study and summarize the tasks and achievements of the Institute.

## Proposed method

### Related work

In geolocation and navigation communities, the requirements for location data are often expressed in terms of the accuracy, integrity, and availability of location information in time and space. The problem lies in these requirements within the scope of logistics applications and the technologies available to meet the needs. Musa considers the diversity of potential applications and the many technologies currently available or under development to meet various industrial needs. The answer to this question is not as simple as it initially seemed. Based on professionally designed Delphi research and experts from a wide range of industries, he provides some specific information and guidance in this area. The results are available for logistics, geolocation and the semiconductor industry to determine the geolocation needs of various applications and how to meet them [11]. Rouquet compared two distribution models

involving experiential logistics: own farms, consumers choosing their own fruits and vegetables, and large retail stores that traditionally sell themselves. Many stores attempt to dramatize low prices by providing extraordinary experience to consumers; the models they choose reveal the concept in a distinctly different way. "Choose you" is an example of a natural staging strategy that encourages consumers to relive the hunter-gatherer experience by combining the product with its accompanying logistics. The two cases he studied show the same desire to develop experiential logistics, but from two opposite perspectives: one based on hedonism and the other based on utilitarianism [12]. Ali bridges the current research gap by developing a model based on extensive empirical evidence that is the interaction between CCLR, resilience and firm performance in the pervasive product supply chain. The hybrid approach is used with qualitative data from interviews and quantitative data from surveys throughout the supply chain. The analysis consists of a contingency theory and a resource-based theory. The survey identified four important CCLR sources and six resources for building resilience. Practical implications the findings will help to improve management understanding of key risk sources in cold chain logistics and to create resources that are indispensable for resilience [13]. Chaudhuri conducted a literature review based on content analysis of 38 selected research papers published between 2000 and 2016 to outline data capture and techniques for collecting and sharing data. It is important to understand how to continuously monitor conditions such as temperature, humidity, vibration, etc. to support real-time quality assessment, determine the actual remaining shelf life of the product, and use it for cold chain decision making. Analysis of such data over a longer period of time can also reveal patterns of product degradation under different transportation conditions, which can result in redesigning the transportation network to minimize quality loss or take precautions to avoid adverse transportation conditions. The results of this study may be beneficial for multiple participants in the cold chain, such as food processing companies, logistics service providers, ports and wholesalers, and retailers to understand how to effectively use data to better serve in the cold chain make a decision [14].

Abushaikha Research has developed an original tool for measuring waste reduction in warehouses and provides insights into the growing field of lean warehousing research. The tools developed provide guidance to logistics managers on how to optimize waste in each warehousing activity. The results also tell logistics managers how to improve distribution performance through lean warehousing. This is the first academic work to reveal the relationship between warehouse waste reduction practices, warehouse operations performance, distribution performance and business performance [15]. Sanjay conceptualizes the consumption of raw materials in road projects as a problem of logistics network allocation. By integrating the three phases of material movement, a linear programming formula with appropriate decision variables is constructed. A series of LP scenes are solved using the LP solver to determine the optimal motion of the aggregates consumed in the layers of the different sections. The results obtained from the model show that logistics using the optimization plan for the entire road project can save logistics costs more than using common sense [16]. Logistics has evolved from a description-based discipline to a theoretical foundation based on other business disciplines to define, interpret, and understand complex interrelationships to identify key areas and key concepts of the discipline. The medium-scale theory enables researchers to focus on these internal operations to gain a deeper understanding of the extent and conditions of the effects of logistics phenomena and the mechanisms by which these results are manifested. Stank proposes an MRT method based on background and mechanism to promote medium-level logistics research, outlines how to guide the process in the middle range, and provides templates and examples for deductive and inductive MRT [17]. Frehe proposes a new concept for the sustainable implementation of crowd logistics services. Follow the design science process to develop the proposed crowd logistics business model concept. Identify four relevant steps

companies should follow to implement sustainable crowd logistics services. It also identified open research issues and guided five research tasks, which may lead to a deeper understanding of this emerging field [18]. Jamal studied the environmental sustainability practices of third-party logistics providers in developing countries and analyzed TPL's efforts to implement green practices through the Moroccan TPL case study. The findings suggest that internal and external drivers have prompted TPL to implement green practices, and internal and external barriers have hampered these practices [19]. Dey discusses the importance of incorporating sustainability into supply chain operations, based on a review of existing literature and then describes various areas of logistics capabilities that enable sustainability. Provides short-term and long-term recommendations for the successful implementation of supply chain logistics functions. The survey results show that the role and importance of logistics is very small in understanding the pursuit of logistics for sustainable development [20].

## Analytic hierarchy process to determine indicator weights

The quantification of indicators requires decisions from relevant business leaders or domain experts with certain experience, but for qualitative indicators, an easy way to quantify is to rank their importance. However, in the past research, most scholars will question the subjectivity of expert evaluation. Therefore, when determining the weight of indicators, it needs to be more scientific and ensure simplicity to meet the operability of enterprises. The analytic hierarchy process has certain aspects in this respect. So this article uses the analytic hierarchy process, combined with expert scoring to empower the indicators [21]:

1. Establishing a hierarchical structure According to the structure of the analytic hierarchy process and the influencing factors of the Internet rural logistics talent innovation mechanism in this study, the influencing factors are stratified, including the target layer, which is the overall goal of the research evaluation, and the objectives of this research have been It is not necessary to be reflected in the table; the criterion layer is the standard for the classification of research indicators, that is, classification from inside and outside the enterprise, listed as a primary indicator in the table; the indicator level is the lowest level, which is the criterion layer. The sub-categories are further classified according to internal and external characteristics, and are listed as secondary indicators in the table. And these indicators are set according to the principle of indicator setting, providing a scientific and reasonable basis for the estimation and evaluation of indicators [22].

2. Construction judgment matrix the judgment matrix of this study is constructed on the basis of the hierarchical structure of the influencing factors of the Internet rural logistics talent innovation mechanism, that is, the evaluation index system. Using the Delphi method, select relevant professionals of the enterprise as the survey object, and obtain the data needed for the research through questionnaire survey [23].

3. Full vector calculation
   According to the method of matrix operation, the influencing factors of each judgment matrix account for the weight value of the upper-level factor, determine the weight vector, and construct the corresponding mathematical model by the feature root method:
   First, the judgment matrix row element product Mj is obtained according to the weight of each index.:

$$M_j = \prod_j^n a_{ij} (i = 1, 2, \cdots n) \tag{1}$$

Second, build Mi's n-th root Wi:

$$W_i = n\sqrt{M_i} \tag{2}$$

Again, the normalized processing vector W = [W1, W2,. . . Wn] specific model is shown below:

$$W_i = \frac{\bar{W}_i}{\prod_j^n W_j} \tag{3}$$

Finally, determine the maximum eigenvalue of the judgment matrix $\lambda_{max}$:

$$\lambda_{max} = \prod_i^n \frac{(AW)_i}{nW_i} \tag{4}$$

4. Consistency test
   The consistency test is used to determine whether the judgment matrix is logical. The determination of the weight value needs to be based on this. Therefore, the judgment matrix of the influencing factors of the Internet rural logistics talent innovation mechanism also needs to be tested for consistency.

5. Total sorting and inspection
   Firstly, the total ranking is determined, that is, the relative weight of each index factor in the method is calculated relative to the first-level index. If only the target layer and the criterion layer, that is, the first-level index, the second-level single sorting can be regarded as the result of the total sorting.

## Construction of fuzzy comprehensive evaluation model

In the above study, it can be determined that the indicator system is three-tier, so it is necessary to evaluate from the single-level and multi-level dimensions in the evaluation. Therefore, in the actual operation, this study expands the fuzzy comprehensive evaluation method and adds a multi-level analysis, that is, constructs a multi-level fuzzy comprehensive evaluation method, which can be verified in the empirical analysis of the following text. The specific contents are as follows:

1. Single-level fuzzy comprehensive evaluation model
   In this model, two finite universes U = {U1, U2,. . .}, V = {V1, V2,. . .} are assumed. The former represents a collection of all evaluation indicators, and the latter represents a collection of all reviews. The result of the single factor evaluation is determined based on the first evaluation factor, that is, the evaluation decision matrix of m evaluation factors can be expressed by the following formula:

$$R = \begin{matrix} R1 \\ \vdots \\ Rm \end{matrix} = \begin{bmatrix} r11 & \cdots & r11 \\ \vdots & \ddots & \vdots \\ r11 & \cdots & r11 \end{bmatrix} \tag{5}$$

According to the above content, this paper mainly compares the importance of evaluation indicators, then calculates and normalizes the weight coefficient, and then weights the

weights with the evaluation factors, and then obtains the fuzzy comprehensive evaluation results of each evaluation index. The specific model is as follows:

$$A \times R = (a_1, a_2, \cdots, a_p) = \begin{bmatrix} r11 & \cdots & r1m \\ \vdots & \ddots & \vdots \\ rp1 & \cdots & rpm \end{bmatrix} = B \tag{6}$$

2. Multi-level fuzzy comprehensive evaluation model

Through a certain attribute, the evaluation factor set U can be divided into m subsets, and then the second level evaluation factor set can be obtained according to the model. The single-level fuzzy comprehensive evaluation method is used to evaluate the evaluation factors in each subset Ui. If the factor weight in Ui is determined as Ai and its judgment matrix is represented by Ri, then the comprehensive evaluation result of the first subset can be used. The following formula:

$$B_i = A_i \times R_i = [b_{i1}, b_{i2} \cdots b_{in}] \tag{7}$$

Construct an evaluation matrix based on the previous comprehensive evaluation of the subset of m evaluation factors in the set U. The specific model is as follows:

$$R = \begin{matrix} B_1 \\ \vdots \\ B_2 \end{matrix} = \begin{bmatrix} b_{11} & \cdots & b_{1n} \\ \vdots & \ddots & \vdots \\ b_{m1} & \cdots & b_{mn} \end{bmatrix} \tag{8}$$

The content of this chapter mainly involves the construction of the evaluation index of the influencing factors of the Internet rural logistics talent innovation mechanism, and uses the multi-level fuzzy comprehensive evaluation method to evaluate, and the specific steps of the evaluation are given in this chapter, so as to provide a basis for the case analysis. A systematic, scientific and perfect system plays a decisive role in the rationality of evaluation. Therefore, the construction of the indicator system should fully consider the characteristics of each indicator and follow certain principles.

$$N_e = \frac{2p_e V_l n}{60\tau} = \frac{p_e V_l n}{30\tau} = K_1 p_e n \tag{9}$$

$$M_1 = \frac{60}{2\pi} \times \frac{N_e}{n} = 9.55 \frac{N_e}{n} = K_2 p_e \tag{10}$$

$$DL = 0.133 \times 57.3 \times \arctan \frac{\sqrt{1 - \left[-\tan\left(\frac{LT}{57.3}\right) \times \tan(SL)\right]^2}}{-\tan\left(\frac{LT}{57.3}\right) \times \tan(SL)} \tag{11}$$

$M_1$ is the output torque of the diesel engine: $K_1 = V_l/30\tau$; $K_2 = 0.955\, K_1$.

If the set of all variables is X, the set of evidence variables is E, and the set of query variables is Q:

$$p(Q|E = e) = \frac{p(Q, E = e)}{p(E = e)} \tag{12}$$

$$d(x_i) = \sum_{i=1}^{n} d(x_i, x_j) \tag{13}$$

Its expression is as follows:

$$\min_i = \min_i - k \times steplen(i) \tag{14}$$

$$\max_i = \max_i + k \times steplen(i) \tag{15}$$

$$K \equiv (K^2/K) = 2 \tag{16}$$

K is the average node degree in the network.

$$P_I(K) = \sum_{K_0 \geq K}^{k} P(K_0)\binom{K_0}{K}(1-H)^K H^{K_0 - K} \tag{17}$$

Using the new degree distribution as in equation:

$$K_1 = K_0 * (1 - H_Y) \tag{18}$$

$$K_1^2 = K_0^2(1 - H_R)^2 + K_0^2 H_R(1 - H_R) \tag{19}$$

Combined with the zero-boundary state criterion of network collapse, we can get:

$$K_1^2/K_0 = K_1^2/K_0(1 - H_R) + H = 2 \tag{20}$$

Llet $K_0 = K_0^2/K_0$ be calculated from the degree distribution in the initial network:

$$\int_{-\infty}^{+\infty} P(K)DK = \int_{-\infty}^{+\infty} CK^{-\lambda}DK = 1 \tag{21}$$

It can be derived from formula:

$$C = (\lambda - 1)m^{\lambda - 1} \tag{22}$$

$K_0$ Is calculated as:

$$K_0 = K_0^2/K_0 = \sum_{KMIN}^{K_{MAX}} K_0^2 PK_0 / \sum_{KMIN}^{K_{MAX}} K_0 PK_0 \tag{23}$$

Putting:

$$PK_0 = CK_0^{-\lambda} = (\lambda - 1)M^{\lambda - 1}K_0^{-\lambda} \tag{24}$$

Into the above formula, we can get:

$$K_0 = \int_{M}^{K_{MAX}} K_0^2(\lambda - 1)M^{\lambda - 1}K_0^{-\lambda}DK_0 / \int_{M}^{K_{MAX}} K_0(\lambda - 1)M^{\lambda - 1}K_0^{-\lambda}DK_0 \tag{25}$$

## Experiments

### Rural logistics talent experimental data

The company is transformed from a traditional logistics company and is the largest logistics express trading network platform in China. Co-investment by giants in the logistics industry, to create a three-dimensional ecological model of the logistics industry, using advanced Internet information technology to develop a set of information to get involved in the logistics industry practitioners The platform connects the main body of logistics demand, can be online transaction, settlement, supervision and evaluation; establishes urban logistics nodes offline, plays the role of collecting goods and distribution, utilizes social transportation capacity, builds national transportation network, promotes the overall development of logistics industry, and takes up the responsibility Promote the social responsibility of the reform and upgrading of the logistics industry. In the road transport industry, there are a large number of small and medium-sized logistics companies, special-line logistics companies, truck transport companies, distribution companies and other small and medium-sized logistics enterprises. In order to dock cargo owners, micro-loan companies, banks, insurance companies and other supporting enterprises, through the integration of information systems, build a new logistics ecosystem to provide practical solutions for solving the problem of small road freight and scattered traffic.

At present, the platform has more than 15,000 members of small and medium-sized micrologistics, which are more than 7000s line members and more than 7,000 lines in the province; a total of 83% of city-level coverage, 82% of district and county level coverage; There are 26 parks and 100 hubs. At present, with strong capital platform and link logistics needs. The main body, the matching network and the members of the line, are committed to a little national transmission, without direct transit. The company's hub has line members, outlet members, and large trucks. It is a distribution and distribution platform. It has a hub to help members reduce money and time costs, and saves effort. The company currently has more than 200 hubs and 26 parks, with the goal of establishing 1,000 transshipment centers. The linkage between hubs and distribution centers can be sent nationwide, with no direct transit. Interconnected hubs improve timeliness and reduce the risk of transit handling. As shown in Table 1, the number and proportion of workers are divided into logistics personnel in a certain area.

The industry and customers have generally relied on the innovative operation and standardized operation of the logistics group since its establishment. The logistics group has also gained good economic and social benefits on this basis, and with many social honors, it has become the first-class in the country. The ranks of logistics companies. Since its establishment, the operating income, profits and taxes paid by the logistics group are shown in Table 2.

### Model simulation setup

As long as the function of implementing the flow chart of the simulation program can be run, any technical means is feasible. This paper uses MATLAB to simulate, transforms the actual

**Table 1. The number and proportion of workers in a certain area of logistics.**

| Job classification | Number of people | The proportion |
|---|---|---|
| Shelf | 15000 | 19.2% |
| Sorting | 10000 | 12.8% |
| Stacking | 8000 | 10.2% |
| Package | 20000 | 25.6% |
| Delivery | 25000 | 32% |

**Table 2. Relevant economic indicators of logistics group (unit: 100 million yuan).**

| Index | 2015 | 2016 | 2017 | 2018 |
|---|---|---|---|---|
| Operating income | 296.89 | 314.29 | 707.5 | 1243.64 |
| Pay taxes | 0.3789 | 1.5549 | 3.777 | 4.83 |
| Profit | 0.8101 | 1.3 | 6 | 10.05 |

system into an applied mathematical model through relationship analysis, and then transforms it into a linear algebra model by using the operational research method. Then use MATLAB to write the program and realize the simulation through matrix operation.

Under the conventional strategy, the distribution team directly selects the route with the largest total logistics demand and delivers it according to each consumer market. In the delay strategy, the market is subdivided into the far end and the near end, and the total distance at the far end of a certain path. When the demand reaches a certain amount, the distribution can be carried out. The simulation can be divided into the following core processes:

1. Initialize each parameter, including the distribution tasks that have not been delivered in the previous cycle, the demand distribution of each cycle, and the distance from each consumer market to the location of the network.

2. Arbitrary demand for this cycle is added to the logistics needs of each consumer market.

3. The routing path selection for this round of cycles is performed according to the rules of the conventional strategy and the delay strategy.

4. Set the number of packages to be loaded by the distribution fleet according to the selected route package.

5. The delivery operation is performed according to the rules of the regular policy and the delay policy.

By simulating the situation of the T-cycle, the total number of parcels completed in the cycle T and the total mileage of the fleet are counted. Finally, the feature quantity 1 = the number of completed packages / the total mileage of the journey is used to characterize the efficiency of the delivery operation. At the same time, the statistical in-transit time t = total travel distance/distribution vehicle speed, the simulation period terminates the simulation when the simulation period reaches T, and the value of the characteristic statistics of the full-cycle simulation is output. The magnitudes of the feature quantities I and t can represent different situations and different the difference in service level and resource utilization of the distribution operation under the strategy determines whether the delivery operation organization plan is better. After multiple simulations, the results need to be evaluated. By averaging and comparing I values under the two strategies of competition and cooperation, if the value of I is larger, it means that the delivery operation efficiency is higher in this case, saving resources and costs. In the market situation, the parameters and market plans can be adjusted, the simulation results and the value of I are compared, and a better strategy is sought to achieve the purpose of optimization.

According to the needs of the project, this study mainly listened to the opinions and suggestions of relevant experts in the logistics industry when determining the indicator system, and determined that the indicator system is feasible. The criterion level, that is, the first level indicator contains 9 influencing factors, and the indicator level is the second level. The indicator is a total of 49 indicators, which serve as the basis and basis for the evaluation.

**Table 3. Comparison of simulation results under actual conditions.**

| Number of simulations | 1 | 2 | 3 | 4 | 5 | 6 | 7 | 8 | 9 | 10 |
|---|---|---|---|---|---|---|---|---|---|---|
| Competitive situation | 2.59 | 2.55 | 2.58 | 2.52 | 2.57 | 2.62 | 2.54 | 2.56 | 2.53 | 2.57 |
| Work efficiency | 4.93 | 4.87 | 4.96 | 4.99 | 5.22 | 5.10 | 4.97 | 5.14 | 4.97 | 4.92 |
| Operating hours | 30.32 | 31.67 | 32.42 | 31.08 | 31.41 | 30.32 | 31.11 | 31.40 | 30.68 | 31.08 |

## Discussion

### Simulation result analysis

According to the collection and research of the above-mentioned documents, as well as the establishment of preliminary models, this paper considers that the external influence factors include national policy orientation factors, technological environment changes, external economic environment changes, social and cultural changes, and customer needs. After the physical parameter setting of the model is completed, 30 cycles and 10 rounds of simulation are performed. Firstly, the distribution operation is compared with the feature quantity of the competition and cooperation situation under the conventional strategy. The distribution operation related indicators are inspected from the assumption that the market share occupied by a single network point in the competition, and the results are separated by slashes in the table. As shown in Table 3.

As shown in Fig 1, it can be seen that after the cooperation, the efficiency of the distribution operation is improved, and the transit time is basically the same, which indicates that the demand change has a negative impact on the distribution service level and the unit operation cost; In the case of competition, cooperation can increase the efficiency of distribution operations by 1.95 times, and when it accounts for 3/4 of the market, it can increase by 1.3 times. It can be seen that the improvement effect achieved by cooperation when the market share of the network is high is smaller. Types of generator faults is shown in Table 4. Unit operation under different confidence levels under traditional model is shown in Table 5. Shipping time is shown in Fig 2. Unit operation is shown in Fig 3.

According to the above table, this is rated by a number of experts, marking the percentage of the number of people in each comment set. After collecting and sorting the data, the following list can be obtained, and the efficiency is high. Therefore, the degree of membership of each commentary is expressed as a percentage, which is 10.9%, 33.8%, 27.3%, 16.3%, and

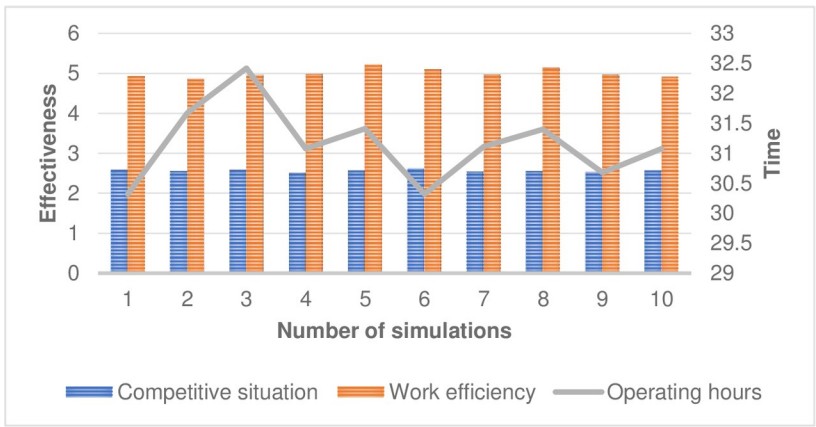

**Fig 1. Curve of comparison of simulation results.**

**Table 4. Types of generator faults.**

| Fault type | Fault category |
|---|---|
| Electrical failure | Stator winding short circuit |
| | Broken stator wire |
| | Stator hydro junction failure |
| Mechanical failure | Rotor unbalance |
| | Air gap eccentricity |
| | Shafting misalignment |

11.7%, respectively, and is multiplied by the score to obtain the score as shown in Table 6 below. Data collection of various users as shown in Table 7. Degree of data loss is show in Fig 4. Lower rating is show in Fig 5.

As shown in Fig 6, according to the above data analysis, it is considered that the comprehensive evaluation score of the influencing factors of the Internet rural logistics talent innovation mechanism is 3.18, the comprehensive ability is generally biased, and the evaluation result is in line with the expected situation. For the managers of the enterprise, the result is a place that needs attention. Combined with the comprehensive evaluation results of the secondary indicators, the indicators with higher weights among the lower scores are rectified and tracked, and the key supervision and each impact are done. Factor analysis and management, compared with previous related research, this method is relatively scientific. According to the weight value, these factors can also be sorted, and some factors can be managed according to the sorting result. The comprehensive membership score is shown in Table 8. Experimental classification results is shown in Table 9. The comprehensive ability evaluation is shown in Fig 7. The result of factor analysis is shown in Fig 8. The result of comprehensive membership analysis is shown in Fig 9.

## Fuzzy comprehensive evaluation analysis

According to the above literature, the hypothesis of the internal influence factors of the Internet rural logistics talent innovation mechanism mainly includes the changes of business operation mode and business strategy, the management advantages of Internet logistics enterprises and the bottlenecks encountered by enterprises. The dimension of the variable is selected and the research model is formed. According to the influence mechanism of internal and external factors, the index system of this paper is proposed. The selection of dimensions and the establishment of the model are based on previous research. Professional and technical personnel in the rural e-commerce industry are indispensable, as shown in Fig 10. Farmers are the direct participants and main beneficiaries of the rural logistics industry. However, in contrast, the peasants in China's netizens are under-educated and their educational level is not ideal. They are accustomed to the traditional mode of production and management, and have poor ability to accept new things. If you don't understand the development of the information society, you can't move forward with the times. Therefore, the improvement of rural logistics talent innovation resistance. Fuzzy comprehensive evaluation analysis is shown in Table 10.

**Table 5. Unit operation under different confidence levels under traditional model.**

| Confidence level | 66.3% | 86.6% | 95.4% | 99.74% |
|---|---|---|---|---|
| Unit operation cost | 539330 | 542990 | 547760 | 555650 |

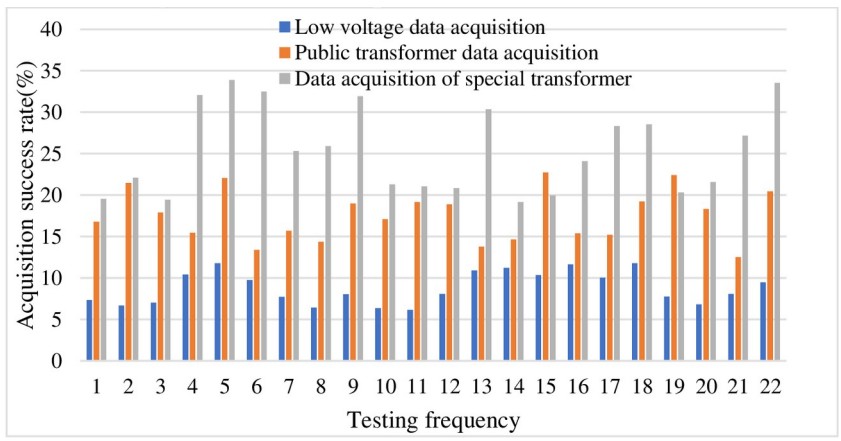

**Fig 2. Shipping time.**

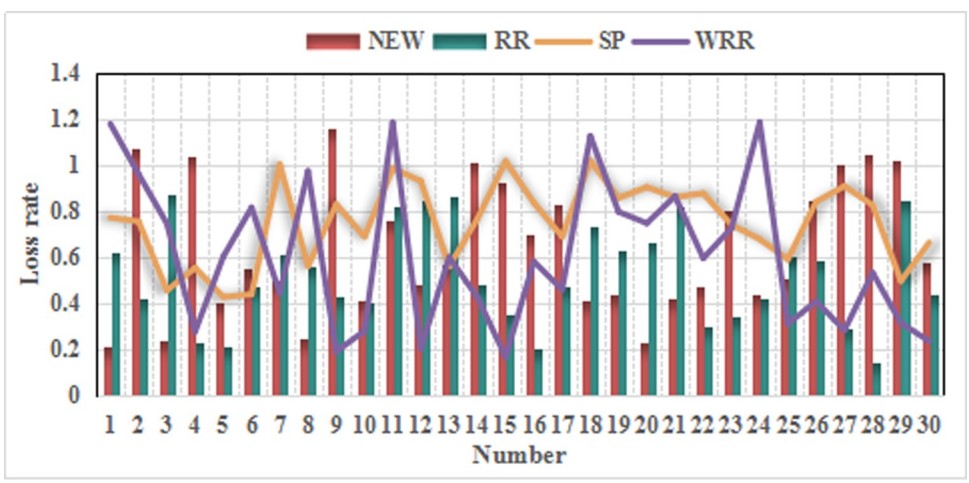

**Fig 3. Unit operation.**

**Table 6. Comparison of simulation results under actual conditions.**

| Comment set | Assignment value | Comment | Membership | Score |
|---|---|---|---|---|
| V5 | 5 | High score | 10.9% | 0.56 |
| V4 | 4 | Higher rating | 33.8% | 1.35 |
| V3 | 3 | Medium rating | 27.3% | 0.82 |
| V2 | 2 | Lower rating | 16.3% | 0.33 |
| V1 | 1 | Low rating | 11.7% | 0.12 |

**Table 7. Data collection of various users.**

| Category | Households(ten thousand) | Acquisition success rate | Application ratio |
|---|---|---|---|
| Low voltage data acquisition | 2084 | >99% | 98% |
| Public transformer data acquisition | 37.65 | >99.5% | 98% |
| Data acquisition of special transformer | 19.84 | >99.5% | 98% |

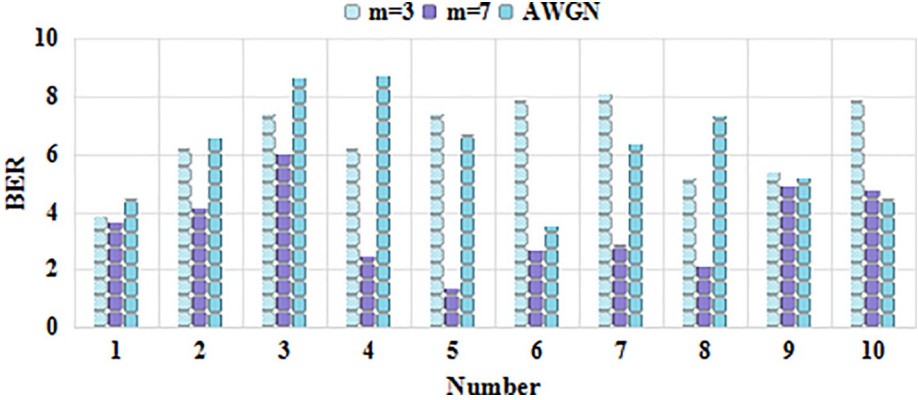

**Fig 4. Higher rating.**

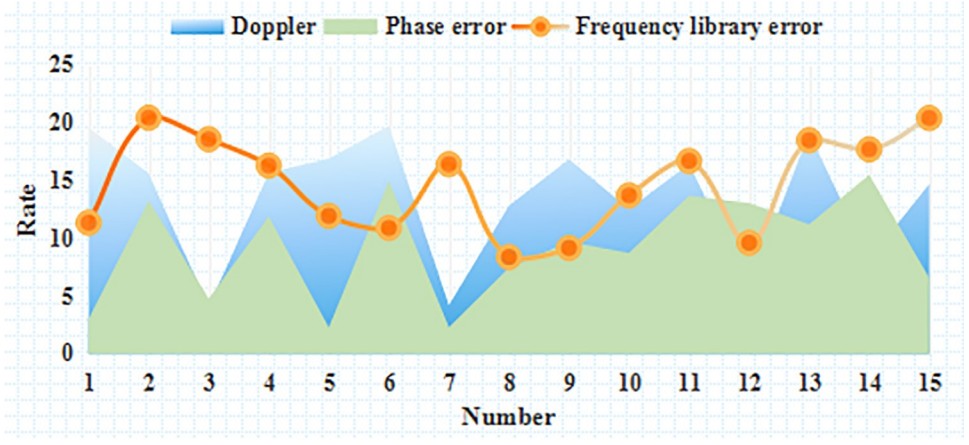

**Fig 5. Lower rating.**

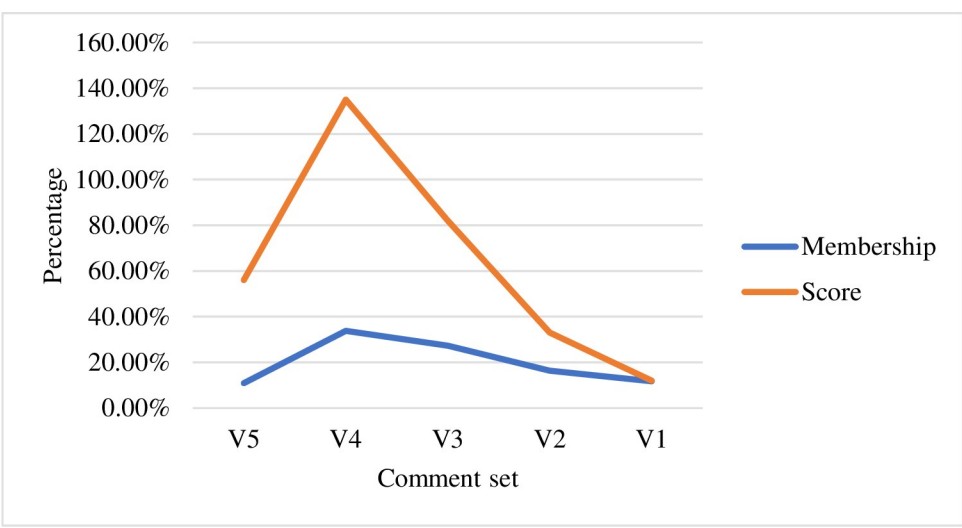

**Fig 6. Model innovation influence factor comprehensive evaluation score curve.**

**Table 8. Comprehensive membership score.**

| Index | V1 | V2 | V3 | V4 | V5 |
|---|---|---|---|---|---|
| Pillar industry | 0 | s>0 | 0.2 | 0.8 | 0 |
| Policy reward | >0 | >0 | 0 | 0.6 | 0 |
| Large-scale strategic deployment | >0 | >0 | 0.4 | 0 | 0 |
| Economic and industrial layout | >0.2 | >0.2 | 0.6 | 0.2 | 0 |
| Reduce logistics inventory | >0 | >0.8 | 0.2 | 0.6 | 0 |
| Information flow replaces logistics | >0 | >0 | 0 | 0.3 | 0.8 |
| Service experience and satisfaction | >0 | >0 | 0.4 | 0.8 | 0 |

**Table 9. Experimental classification results.**

| Algorithm | Classification | Accuracy | Recall | F1 value |
|---|---|---|---|---|
| KNN | high-risk | 78.2% | 81.6% | 75.7% |
| | Low risk | 64.8% | 63.6% | |
| | invalid | 79.0% | 82.7% | |
| | crack | 78.6% | 82.9% | |
| SVM | high-risk | 86.6% | 87.8% | 86.3% |
| | Low risk | 89.2% | 88.6% | |
| | invalid | 86.8% | 90.4% | |
| IOT approach | high-risk | 98.0% | 97.3% | 96.4% |
| | Low risk | 94.9% | 96.6% | |
| | invalid | 97.9% | 95.2% | |

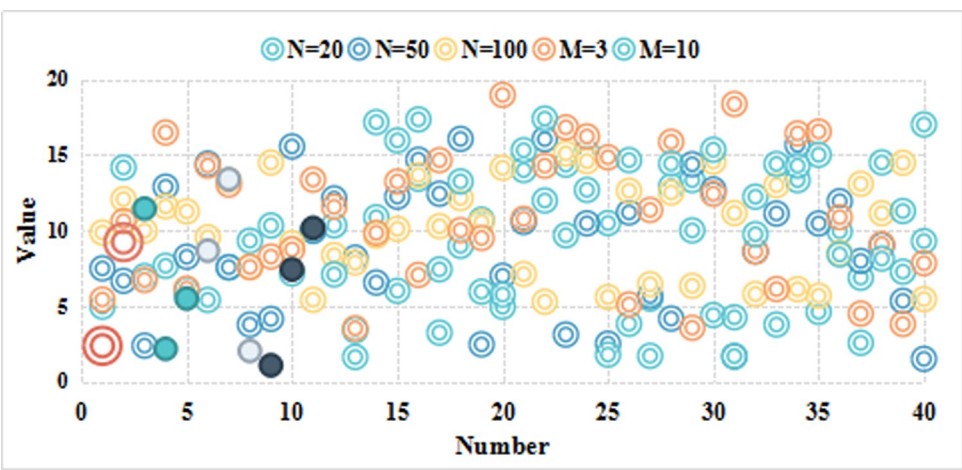

**Fig 7. The comprehensive ability evaluation.**

Information technology is one of the main drivers. As a representative of information technology, the Internet is becoming more and more popular in people's lives, and the mobile Internet continues to penetrate into all aspects of people's lives. People's ability to use information technology, big data statistical analysis, cloud computing, Internet of Things, satellite positioning and other technologies. It is also constantly strengthening. For example, combining the traditional logistics industry with the Internet can well solve the waste of logistics capacity resources. Through intensive management and intelligent logistics park management,

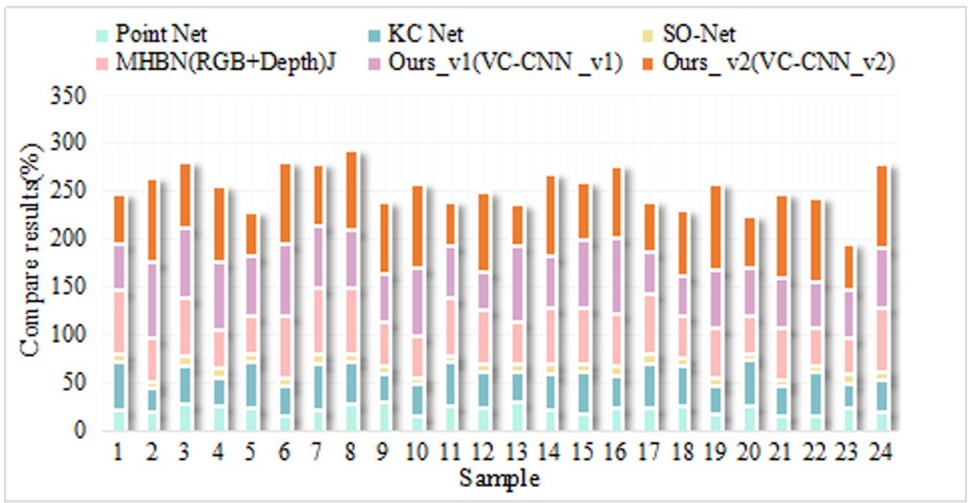

**Fig 8. The result of factor analysis.**

the logistics cost is greatly reduced, the efficiency of logistics operations is improved, and the efficiency is improved. The development of mobile Internet has also made logistics information transmission more efficient and accurate. After the physical parameter setting of the model is completed, 30 cycles and 10 rounds of simulation are performed. In the case of cooperation, compare the feature quantity of the conventional strategy and the delay strategy to evaluate the evaluation of the delivery operation, as shown in Fig 11. The logistics cost is shown in Fig 12. The efficiency of logistics operations is shown in Fig 13.

The lack of long-term talents in the logistics industry and the lack of professional quality of logistics personnel, the lack of resources for logistics enterprises in China's small and medium-sized cities and rural areas, is also the bottleneck problem of the innovation mechanism of Internet rural logistics talents.

These problems also force the transformation of the logistics industry to achieve better development. In summary, the case analysis fully shows that this paper simulates based on the

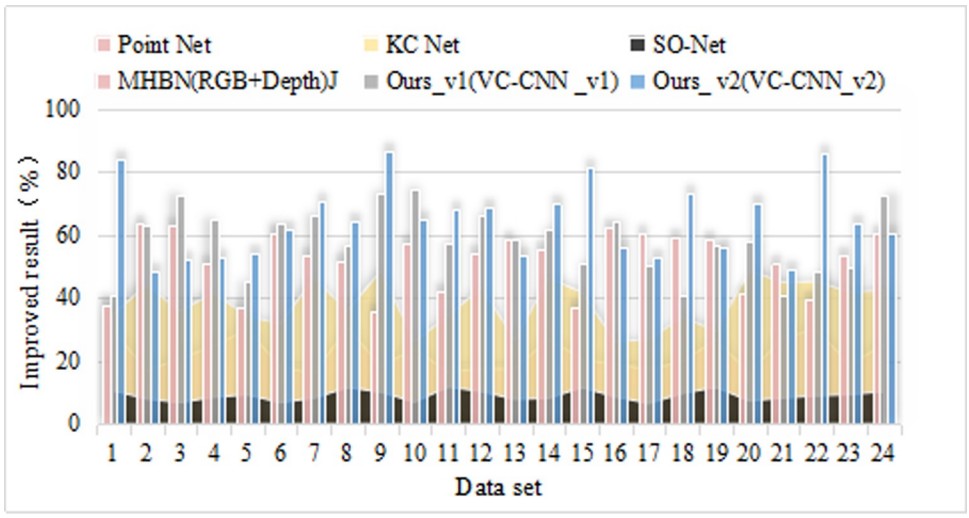

**Fig 9. The result of comprehensive membership analysis.**

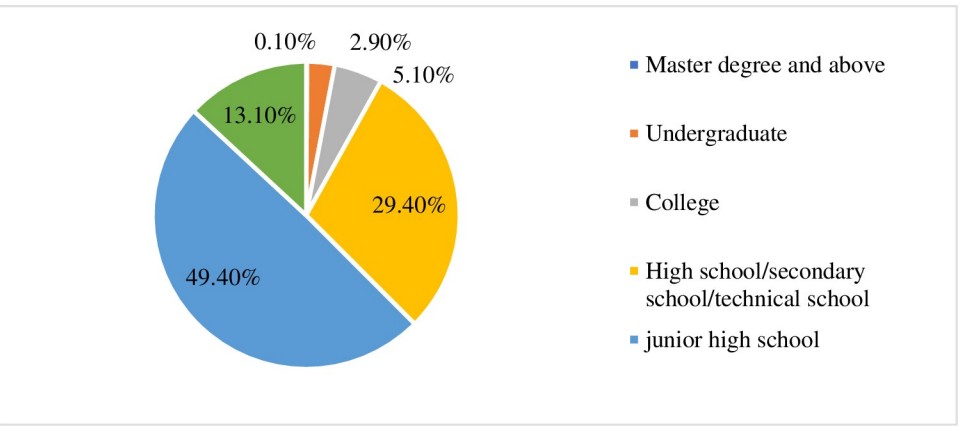

**Fig 10. Rural logistics personnel education ratio.**

**Table 10. Fuzzy comprehensive evaluation analysis.**

| Cooling system failure | Leakage of stator winding |
|---|---|
| | Local blockage of ventilation duct |

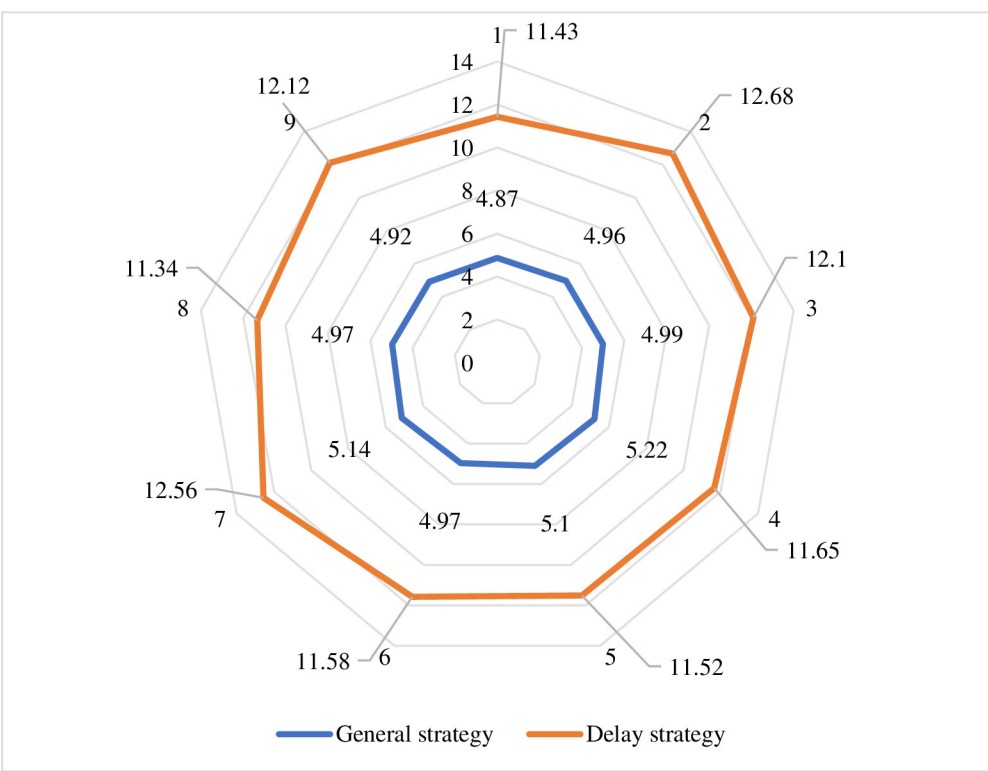

**Fig 11. Comparison of simulation results under conventional strategy and delay strategy.**

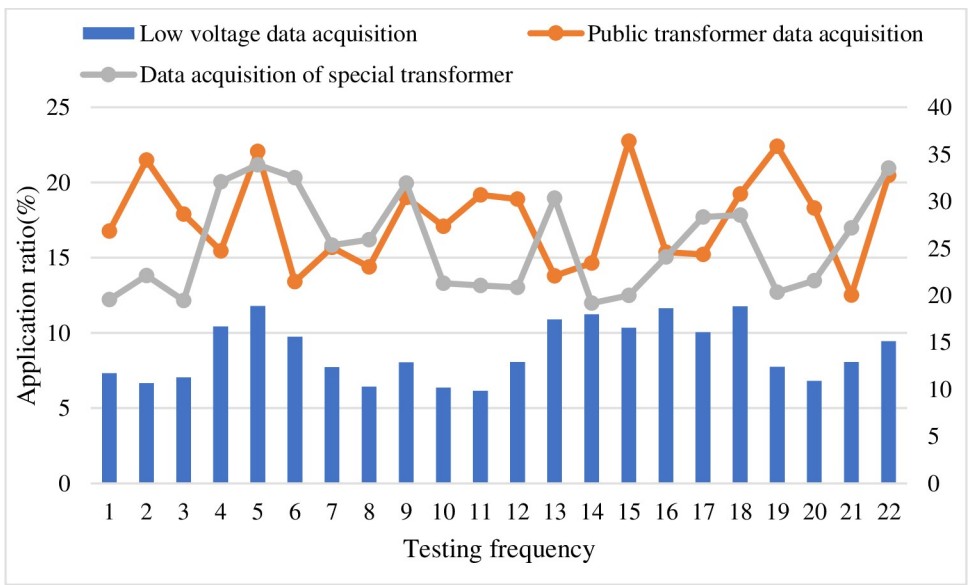

**Fig 12. The logistics cost.**

actual factors such as the segmentation and spatial distribution characteristics of the rural con-
sumer market, and finds that the network can cooperate in the distribution operation and
adopt the delayed delivery strategy in cooperation. To ensure the cost of resources and
improve the efficiency of operations under certain delivery service levels, it shows that cooper-
ation has more advantages than competition, and can further utilize the concentrated effect to
achieve optimization on the basis of cooperation; at the same time, the model assumptions and
structures are close to the actual situation and have practicality. The model simulation results
are relatively clear, and the statistical indicators are intuitive. It can provide theoretical guid-
ance for solving the problems of low efficiency, large demand changes and high cost for the
logistics outlets in the rural e-commerce market. The efficiency of logistics outlets in the rural
e-commerce market is shown in Table 11.

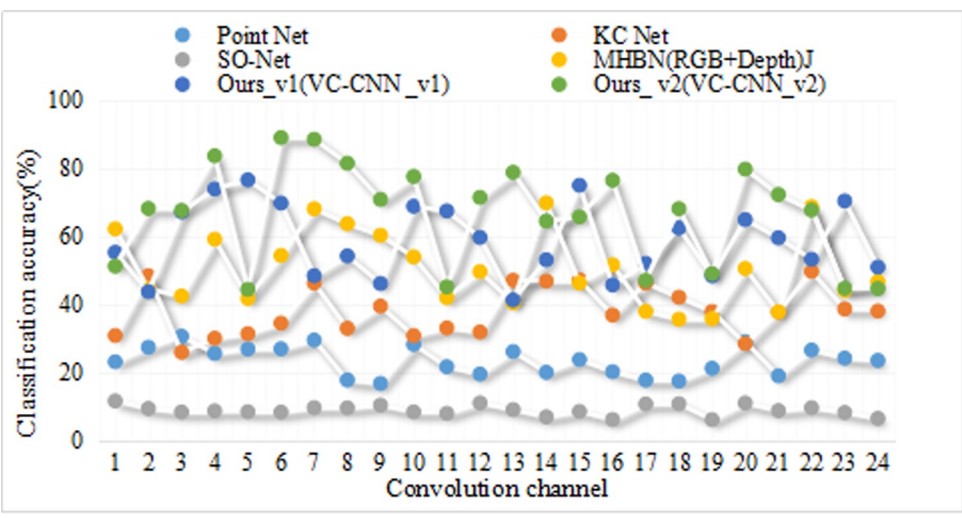

**Fig 13. The efficiency of logistics operations.**

**Table 11.  The efficiency of logistics outlets in the rural e-commerce market.**

| Number of people / Group | Above 130 points | 110–130 points | 100–110 points | 90–100 points | Less than 90 points | Average |
|---|---|---|---|---|---|---|
| A1 | 10 | 20 | 15 | 7 | 3 | 115.65 |
| A2 | 6 | 15 | 16 | 11 | 6 | 104.78 |
| B1 | 9 | 19 | 12 | 10 | 4 | 109.65 |
| B2 | 5 | 15 | 21 | 10 | 5 | 102.45 |

The innovation model needs to be continuously optimized and adjusted as the environment changes. No innovation model is permanent and always running efficiently and efficiently. If the social environment changes, the company should make timely adjustments based on these changes to avoid being Social and market are eliminated. Therefore, logistics companies can only keep up with the times by taking advantage of their own management advantages, combining the development trend of the Internet, advancing with the times, and making bold innovations in business models, business strategies and business models.

## Conclusions

Business model strategy, scientific and technological progress, industrial environment change, customer demand change and other factors have played a more dominant role in logistics industry innovation, and played a more important role in all factors.

Macroeconomic changes, social and cultural factors and their own management advantages also have a certain impact on the innovation of logistics talents, but the importance is lower; while the changes in national policies, layout and bottlenecks are relatively small, although they will also affect talent innovation. But the effect is smaller.

According to the actual situation of rural e-commerce, from the perspective of the whole link, the relationship between the logistics channel and the end transportation of rural e-commerce and the hierarchical relationship of each entity are clarified, and the problems faced by distribution in rural areas are pointed out. And propose innovative solutions to these problems.

## Author Contributions

**Writing – original draft:** Hui Zhan, Xin Zhang, Haiwen Wang.

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
