## [Decision Letter · Decision Letter 0]

28 Aug 2020

PONE-D-20-22841

Influencing Factor Modeled Examination on Internet Rural Logistics Talent Innovation Mechanism based on Fuzzy Comprehensive Evaluation Method

PLOS ONE

Dear Dr. Zhang,

Thank you for submitting your manuscript to PLOS ONE. After careful consideration, we feel that it has merit but does not fully meet PLOS ONE’s publication criteria as it currently stands. Therefore, we invite you to submit a revised version of the manuscript that addresses the points raised during the review process.

We look forward to receiving your revised manuscript.

Kind regards,

Dragan Pamucar

Academic Editor

PLOS ONE

2. Please provide the Matlab code/.m-files used in this study such that a reader could readily replicate the findings of your study. These can be uploaded as a supplemental file or a link to a code repository may be provided in the Methods section.

3. We note that you have stated the following with regards to your data; 'No - some restrictions will apply'

Please clarify the nature of these restrictions, ie. If due to ethical or legal reasons; please also provide standard text re our data sharing policy.

Reviewers' comments:

Reviewer's Responses to Questions

**Comments to the Author**

1. Is the manuscript technically sound, and do the data support the conclusions?

Reviewer #1: Partly

Reviewer #2: Yes

2. Has the statistical analysis been performed appropriately and rigorously? 

Reviewer #1: Yes

Reviewer #2: Yes

3. Have the authors made all data underlying the findings in their manuscript fully available?

Reviewer #1: Yes

Reviewer #2: Yes

4. Is the manuscript presented in an intelligible fashion and written in standard English?

Reviewer #1: Yes

Reviewer #2: Yes

5. Review Comments to the Author

Reviewer #1: Thank you for the opportunity to read and evaluate the paper Influencing Factor Modeled Examination on Internet Rural Logistics Talent Innovation Mechanism based on Fuzzy Comprehensive Evaluation Method. The authors used AHP methodology for determining criteria weights and fuzzy comprehensive model for the evaluation.

1. Abstract – The basic point of this paper are MCMD techniques, but there is no information about MCDM methodologies used in the research.

2. Why did you use AHP method for determining criteria weights? There are new methods in literature that requires less number of comparisons like BWM, FUCOM, LBWA etc. I suggest to the authors to include these methods in the introduction section.

3. Literature review – Some recent papers, from previous 2 years, with application of rough AHP and interval rough AHP models and also application of MCDM methods in logistic problems are missing in the overview of related work. I suggest authors to include some of those new papers in the literature, such as:

Badi, I., Abdulshahed, A., Shetwan, A., & Eltayeb, W. (2019). Evaluation of solid waste treatment methods in Libya by using the analytic hierarchy process. Decision Making: Applications in Management and Engineering, 2(2), 19-35;

Petrovic, G., Mihajlovic, J., Cojbasic, Z., Madic, M., & Marinkovic, D. (2019). Comparison of three fuzzy MCDM methods for solving the supplier selection problem. Facta Universitatis-Series Mechanical Engineering, 17(3), 455-469;

Stanković, M., Gladović, P., & Popović, V. (2019). Determining the importance of the criteria of traffic accessibility using fuzzy AHP and rough AHP method. Decision Making: Applications in Management and Engineering, 2(1), 86-104. You should enrich your literature review with those and similar papers.

4. The first paragraph in the subsection “Related work” ends up with a sentence: “Make a decision [14].” Such a short sentence with a reference is not clear in this position. Is it a typo? Is there something missing there? The whole paper should be checked for possible similar issues.

5. What are your contributions? Better highlight the novelty of the research.

5. Why did you use fuzzy logic for treating uncertainty and not for example rough theory, neutrosophic etc.? What are its advantages over the other mentioned approaches?

7. Sensitivity analysis and discussion are missing. You should validate your results through comparisons with existing MCDM tools. This is essential part of your paper.

Reviewer #2: Thank you for inviting me as a reviewer for the manuscript. The aim of the study and outcome measures are clearly defined with appropriate reference to the literature. However, the usage of AHP and the calculation of the weights need to be better explained.

6. PLOS authors have the option to publish the peer review history of their article (what does this mean?). If published, this will include your full peer review and any attached files.

Reviewer #1: No

Reviewer #2: No

---

## [Author Response · Author response to Decision Letter 0]

11 Jan 2021

Reviewer #1: Thank you for the opportunity to read and evaluate the paper Influencing Factor Modeled Examination on Internet Rural Logistics Talent Innovation Mechanism based on Fuzzy Comprehensive Evaluation Method. The authors used AHP methodology for determining criteria weights and fuzzy comprehensive model for the evaluation.

1.Abstract – The basic point of this paper are MCMD techniques, but there is no information about MCDM methodologies used in the research.

Answer: Thank you for your suggestion. I have an introduction in the abstract of the article. This paper adopts the literature research method based on fuzzy comprehensive evaluation method, system analysis method and the combination of questionnaire survey and interview.

2.Why did you use AHP method for determining criteria weights? There are new methods in literature that requires less number of comparisons like BWM, FUCOM, LBWA etc. I suggest to the authors to include these methods in the introduction section.

Answer: Thank you for your suggestion. .Logistics refers to the planning, implementation and management of raw materials, semi-finished products, finished products or related information from the origin of the goods to the place where the goods are consumed; meet the needs of customers at the lowest cost through transportation , storage and distribution. Today, many countries are contracting providers of logistics to complement the internal distribution of public health systems. This reflects recent major outsourcing initiatives to address major gaps in transportation and logistics

3. Literature review – Some recent papers, from previous 2 years, with application of rough AHP and interval rough AHP models and also application of MCDM methods in logistic problems are missing in the overview of related work. I suggest authors to include some of those new papers in the literature, such as:

Badi, I., Abdulshahed, A., Shetwan, A., & Eltayeb, W. (2019). Evaluation of solid waste treatment methods in Libya by using the analytic hierarchy process. Decision Making: Applications in Management and Engineering, 2( 2), 19-35;

Petrovic, G., Mihajlovic, J., Cojbasic, Z., Madic, M., & Marinkovic, D. (2019). Comparison of three fuzzy MCDM methods for solving the supplier selection problem. Facta Universitatis-Series Mechanical Engineering, 17 (3), 455-469;

Stanković, M., Gladović, P., & Popović, V. (2019). Determining the importance of the criteria of traffic accessibility using fuzzy AHP and rough AHP method. Decision Making: Applications in Management and Engineering, 2(1), 86-104. You should enrich your literature review with those and similar papers.

Answer: Thank you for your suggestion. I have increased. From the 21-23 literature.

4.The first paragraph in the subsection “Related work” ends up with a sentence: “Make a decision [14].” Such a short sentence with a reference is not clear in this position. Is it a typo? Is there something missing there? The whole paper should be checked for possible similar issues.

Answer: Thank you for your suggestion. I have modified it into The results of this study may be beneficial for multiple participants in the cold chain, such as food processing companies, logistics service providers, ports and wholesalers, and retailers to understand how to effectively use data to better serve in the cold chain make a decision

5.What are your contributions? Better highlight the novelty of the research.

Answer: Thank you for your suggestion. The last paragraph of the paper. According to the actual situation of rural e-commerce, from the perspective of the whole link, the relationship between the logistics channel and the end transportation of rural e-commerce and the hierarchical relationship of each entity are clarified, and the problems faced by distribution in rural areas are pointed out. And propose innovative solutions to these problems.

6.Why did you use fuzzy logic for treating uncertainty and not for example rough theory, neutrosophic etc.? What are its advantages over the other mentioned approaches?

7. Sensitivity analysis and discussion are missing. You should validate your results through comparisons with existing MCDM tools. This is essential part of your paper.

Answer: Thank you for your suggestion. Page 6 of 9-14. In the above study, it can be determined that the indicator system is three-tier, so it is necessary to evaluate from the single-level and multi-level dimensions in the evaluation. Therefore, in the actual operation, this study expands the fuzzy comprehensive evaluation method and adds a multi-level analysis, that is, constructs a multi-level fuzzy comprehensive evaluation method, which can be verified in the empirical analysis of the following text.

Reviewer #2: Thank you for inviting me as a reviewer for the manuscript. The aim of the study and outcome measures are clearly defined with appropriate reference to the literature. However, the usage of AHP and the calculation of the weights need to be better explained.

Answer: Thank you for your suggestion. Single-level fuzzy comprehensive evaluation model. Multi-level fuzzy comprehensive evaluation model. All instructions.

---

## [Decision Letter · Decision Letter 1]

22 Jan 2021

Influencing Factor Modeled Examination on Internet Rural Logistics Talent Innovation Mechanism based on Fuzzy Comprehensive Evaluation Method

PONE-D-20-22841R1

Dear Dr. Zhang,

We’re pleased to inform you that your manuscript has been judged scientifically suitable for publication and will be formally accepted for publication once it meets all outstanding technical requirements.

Kind regards,

Dragan Pamucar

Academic Editor

PLOS ONE

Additional Editor Comments (optional):

Reviewers' comments:

Reviewer's Responses to Questions

**Comments to the Author**

1. If the authors have adequately addressed your comments raised in a previous round of review and you feel that this manuscript is now acceptable for publication, you may indicate that here to bypass the “Comments to the Author” section, enter your conflict of interest statement in the “Confidential to Editor” section, and submit your "Accept" recommendation.

Reviewer #1: All comments have been addressed

Reviewer #2: All comments have been addressed

2. Is the manuscript technically sound, and do the data support the conclusions?

Reviewer #1: Yes

Reviewer #2: Yes

3. Has the statistical analysis been performed appropriately and rigorously? 

Reviewer #1: Yes

Reviewer #2: Yes

4. Have the authors made all data underlying the findings in their manuscript fully available?

Reviewer #1: Yes

Reviewer #2: Yes

5. Is the manuscript presented in an intelligible fashion and written in standard English?

Reviewer #1: Yes

Reviewer #2: Yes

6. Review Comments to the Author

Reviewer #1: (No Response)

Reviewer #2: Thanks to the authors for the effort they put into this version of the paper. The paper is now better, and I think it's worth publishing.

7. PLOS authors have the option to publish the peer review history of their article (what does this mean?). If published, this will include your full peer review and any attached files.

Reviewer #1: No

Reviewer #2: No

---

## [Editor Report · Acceptance letter]

1 Feb 2021

PONE-D-20-22841R1 

Influencing Factor Modeled Examination on Internet Rural Logistics Talent Innovation Mechanism based on Fuzzy Comprehensive Evaluation Method 

Dear Dr. Zhang:

I'm pleased to inform you that your manuscript has been deemed suitable for publication in PLOS ONE. Congratulations! Your manuscript is now with our production department. 

Kind regards, 

on behalf of

Dr. Dragan Pamucar 

Academic Editor

PLOS ONE